# Ovarian Accumulation of Nanoemulsions: Impact of Mice Age and Particle Size

**DOI:** 10.3390/ijms22158283

**Published:** 2021-07-31

**Authors:** Eike Folker Busmann, Julia Kollan, Karsten Mäder, Henrike Lucas

**Affiliations:** Department of Pharmaceutical Technology and Biopharmaceutics, Faculty of Natural Sciences 1—Biosciences, Institute of Pharmacy, Martin-Luther-University Halle-Wittenberg, 06120 Halle, Germany; eike.busmann@pharmazie.uni-halle.de (E.F.B.); julia.kollan@pharmazie.uni-halle.de (J.K.); karsten.maeder@pharmazie.uni-halle.de (K.M.)

**Keywords:** nanotoxicology, ovaries, nanoemulsion, biodistribution, optical imaging, accumulation, reproductive aging

## Abstract

Nanotechnology in the field of drug delivery comes with great benefits due to the unique physicochemical properties of newly developed nanocarriers. However, they may come as well with severe toxicological side effects because of unwanted accumulation in organs outside of their targeted site of actions. Several studies showed an unintended accumulation of various nanocarriers in female sex organs, especially in the ovaries. Some led to inflammation, fibrosis, or decreasing follicle numbers. However, none of these studies investigated ovarian accumulation in context to both reproductive aging and particle size. Besides the influences of particle size, the biodistribution profile may be altered as well by reproductive aging because of reduced capacities of the reticuloendothelial system (RES), changes in sex steroid hormone levels as well as altering ovarian stromal blood flow. This systematic investigation of the biodistribution of intravenously (i.v) injected nanoemulsions revealed significant dependencies on the two parameters particle size and age starting from juvenile prepubescent to senescent mice. Using fluorescent in vivo and ex vivo imaging, prepubescent mice showed nearly no accumulation of nanoemulsion in their uteri and ovaries, but high accumulations in the organs of the RES liver and spleen independently of the particle size. In fertile adult mice, the accumulation increased significantly in the ovaries with an increased particle size of the nanoemulsions by nearly doubling the portion of the average radiant efficiency (PARE) to ~10% of the total measured signal of all excised organs. With reproductive aging and hence loss of fertility in senescent mice, the accumulation decreased again to moderate levels, again independently of the particle size. In conclusion, the ovarian accumulation of these nanocarriers depended on both the age plus the particle size during maturity.

## 1. Introduction

Nanosized drug-delivery systems (DDS) are increasingly used as drug carriers, diagnostics, or theranostic agents. Due to their very small size, their physicochemical properties might differ substantially from the properties of micron-sized or larger sized (bulk) materials. For example, they possess increased uptake rates and higher interaction with biological tissue, improved targeting to their site of action, or increased reactivity due to their significantly higher specific surface area. For this reason, nanotechnology in drug delivery often surpasses the kinetics, absorption, distribution, and metabolism properties in comparison to congeneric bulk materials. However, admittedly, all these properties can as well lead to concerns about nanotoxicological side effects [1,2,3,4].

The particle size and shape of nanomaterials play an important role in the fate and interaction of nanosized particles with biological tissue, affecting the adhesion on the cell surface/membrane, the mode of cellular uptake or direct penetration through the membrane as well as the processing of the nanoparticles by endocytic pathways [3,5]. Withal, the smaller the size, the more rapidly and easily nanoparticles can penetrate capillary walls. With a particle size of less than 50 nm, they can enter cells and those smaller than 20 nm are even more rapidly taken from the circulating bloodstream into cells out of blood vessels, exhibiting a similar size of biomolecules such as viruses, proteins, enzymes, and receptors [6,7]. Various nanosized DDS such as solid nanoparticles, lipid nanocapsules, and nanoemulsions of various particle sizes were found to accumulate in the tissue of reproductive organs of rodents such as rats and mice [2,4,7,8,9,10,11]. On the one hand, the accumulation in the ovaries may be desired for the treatment of ovarian dysfunctions and diseases such as ovarian cancers. For example, recent in vitro studies showed induced cellular apoptosis of human ovarian carcinoma cells OVCAR 8 using tocotrienol and curcumin-loaded nanoemulsions [12]. Other studies suggested constant parenteral low dose hormone infusion mimicking the daily hormonal ovarian production throughout the menstrual cycle for the therapy of primary ovarian insufficiency, overcoming the first-pass effect of oral DDS as well as reducing breast cancer risk by decreasing the necessary dose [13,14]. Metformin hydrochloride-loaded nanoparticles were successfully i.v. administered in mice causing no adverse effects on the major organs [15], which may be used as an administration route for the treatment of polycystic ovary syndrome to increase insulin sensitivity and overcome insulin resistance in the ovaries [16,17]. On the other hand, the accumulation of nanosized DDS in the ovaries may exhibit high toxicity risk. Besides the unintended release of active pharmaceutical ingredients through an unwanted ovarian accumulation of the DDS, some nanoparticles were reported to cause toxic harms already by themselves: Some led to inflammation, congestion, extravasations of red blood cells, fibrosis, and apoptosis in ovarian cells of rats [2,8] and others decreased the number of follicles in the ovaries after i.v. application in female mice [2,9].

However, none of these studies investigated both the influence of particle size and aging effects on the accumulation of nanoparticles in ovarian tissue. Therefore, aging alters the capacity of the RES involving multiple organs such as liver, spleen, lymph nodes, and bone marrow in humans as well as in mice [18,19,20] and therefore might change the distribution profile of nanosized DDS over aging. Furthermore, the change of sex steroid hormone levels plays a major role during the aging of females: inducing puberty, regulate the estrous cycle during maturity, and leading to menopause, gradually decreasing the fertility of women over 2–7 years until its complete loss. During that period, the ovarian hormone level and the ovarian stromal blood flow decline significantly [21,22]. Up to today, no single mouse strain has been reported exhibiting all the same reproductive aging effects as human women but still exhibits several similarities [21,23]. At the age of 4–6 weeks, progesterone and estradiol levels increase significantly in mouse serum with the onset of puberty and the first ovulation can be detected [24,25,26]. Adult mice then undergo repeatedly the estrous cycle predominantly within 4–5 days, including the cycling of the ovarian hormone levels [27,28,29]. During reproductive aging, mice however do not undergo complete menopause as humans do, but depending on the mouse strain, their fertility declines after 6–7 months with declining litter size and then ceases nearly completely after 9–12 months of age. During that period, the estrous cycle becomes irregular until complete termination, and therefore the steroid hormone cycle ceases with a decline of estradiol and progesterone level [21,23,28,29,30,31,32,33]. Hence the age and therewith the state of sex steroid hormones and ovarian stromal blood flow may play a major role in the accumulation of nanosized DDS in developing, mature, and aging sex organs of females.

For this study, previously investigated phase inversion-based produced nanoemulsions of medium-chain triglyceride (MCT) were chosen as a suitable formulation for the investigation of their biodistribution in vivo. These nanoemulsions turned out to be easily tunable in particle size from 16 to 175 nm at remarkably narrow distributions by solely changing their lipid:surfactant ratio. As illustrated in schematic Figure 1, it was possible to load these nanoemulsions with the near-infrared fluorescent dye DiR (DiLC18(7)) as a label for in vivo fluorescence imaging [34]. Choosing a near-infrared fluorescent dye as a label allowed probing significantly larger tissue depths since various cell components in the tissue absorb light in ultraviolet and visible wavelengths limiting light penetration to a few hundred microns [35]. For further improvement of the in vivo fluorescent imaging, the immunocompetent hairless SKH1-*Hr^hr^* Elite mouse strain was chosen, which lose their hair within 3 weeks after birth excluding interferences between fur and emitted fluorescence light [36,37]. The chosen dialkylcarbocyanine DiR was already deployed for in vivo imaging and was reported as a non-irritant [38,39,40,41,42]. Because of its long C18 alkyl chains, DiR is highly lipophilic and insoluble in water with a reported logP of 17.4 [43], thus a transfer of such a lipophilic dye out of the lipid nanoemulsion core is highly unlikely, which was described for the similar fluorescent dialkylcarbocyanine dye DiLC18(3) at 37 °C with less than 5% over several weeks for lipid nanoemulsions [44]. However, caution should be taken, when using the near-infrared fluorescent dye DiR for the labeling of DDSs which develop an acidic microenvironment, since DiR might be degraded at pH values below 4 to fluorescent molecules with different properties [45].

## 2. Results and Discussion

### 2.1. Physicochemical Properties of the Nanoemulsions

Particle size analysis, zeta potential, osmolality, and pH measurements were performed to obtain an overview of the physicochemical properties of the four different nanoemulsion formulations. Hydrodynamic particle diameters were determined by dynamic light scattering and expressed as the commonly used Z average (z_ave_), which is defined as the scattered light intensity-weighed harmonic mean diameter. Furthermore, the width of the overall particle size distribution was expressed as the dimensionless and commonly used polydispersity index (PDI) (ISO 22412 and [46]). Both calculated mean values of three individually produced batches are shown in Figure 2 for the four different sized nanoemulsions. Therefore, the targeted particle sizes of 25, 50, 100, and 150 nm in droplet diameter were reproducibly achieved for the nanoemulsions NE25 (orange bars) with 25.7 nm and NE50 (red bars) with 50.5 nm z_ave_ with very low PDI values of 0.028 and 0.035, indicating a very narrow distributed monomodal size distribution. The nanoemulsions NE100 (blue bars) with 97.7 nm and NE150 (green bars) with 144.9 nm z_ave_ were slightly below the targeted droplet diameter. Both respective PDI values of 0.086 and 0.146 indicate still narrow size distributions. Furthermore, the electrokinetic potentials at the slipping plane of the nanoemulsion droplets were determined as zeta potential by electrophoretic light scattering in 0.1x phosphate-buffered saline (PBS) at pH 7.4. The zeta potentials of the four different nanoemulsions were nearly neutral between −3.15 to −1.98 mV.

The osmolalities of the nanoemulsions were measured by cryoscopy. They were isotonic for the nanoemulsions NE100 and NE150 and nearly isotonic for the nanoemulsions NE25 and NE50 so that a painless i.v. application without any vascular damage was ensured [47,48]. With a decreasing pH value from 5.6 to 4.0 for increasing particle size, the nanoemulsions were slightly acidic and therewith lied as well in an acceptable range for small volume i.v. injections between pH 3–11 [47,49]. Furthermore, within this measured pH range of the nanoemulsions and for the short period between production one or two days before i.v. application until the end of the animal trials one day after application, an acid-induced degradation of the incorporated fluorescent dye DiR was very unlikely, which was described as chemically stable for pH values above 4.0 [45].

### 2.2. Cytotoxic Properties of the Nanoemulsions

For the determination of the cytotoxic properties of the four different sized nanoemulsions, cellular toxicity assays by resazurin reduction were performed with mouse embryonic fibroblasts (3T3) and normal human dermal fibroblasts (NHDF) after 24 h of incubation with the nanoemulsions. The resulting dose-response curves are shown in the Appendix A. The median inhibitory concentration (IC50), at which the cells possess half the viability of untreated cells, is displayed in Figure 3 for both cell lines.

For both investigated cell lines, the IC50 value increased up to five times from the NE25 to the NE150 nanoemulsion, indicating decreasing cytotoxicity with increasing particle size of the nanoemulsion. Overall, the 3T3 fibroblasts reacted more sensitively to the nanoemulsions than the NHDF fibroblasts, with significantly higher IC50 values. The lower cell viability with decreasing particle size can be explained by the increasing amount of the surfactant Kolliphor^®^ HS 15 to achieve smaller particle sizes. This non-ionic surfactant was already detected in previous research as the main cell viability inhibiting ingredient [34,50]. Therefore, it was likely that insoluble 12-hydroxystearic acid as a metabolic degradation product of Kolliphor^®^ HS 15 formed needle-like crystals and thus caused cell death in vitro for both cell lines [34].

### 2.3. Nanoemulsion Interaction with the Blood Cells

To investigate the interaction of the DiR fluorescent dye-loaded nanoemulsions with blood cells, a volume of 50 µL of each nanoemulsion was incubated for 4 h in 100 µL whole mouse blood. The average radiant efficiency (ARE) of the fluorescence signal was determined of the separated plasma and blood cells by fluorescence imaging. Their calculated PAREs for both separated fractions are displayed in Figure 4. For the nanoemulsions NE50, NE100, and NE150, solely 0.3% to 0.6% PARE was detected in the fraction with the blood cells, indicating that nearly no nanoemulsion droplets underwent a cellular uptake into the blood cells, nor fluorescent dye was transferred from the lipid core into the blood cells. For the smaller-sized nanoemulsion NE25 1.3% PARE was detected in the blood cell fraction. Hence, either solely a marginal amount of nanoemulsion droplets entered the blood cells through cellular uptake or transferred the fluorescent dye DiR into the blood cells, due to a highly increased specific surface area of smaller-sized nanoemulsion droplets available for such a transfer. Overall, all four nanoemulsions and their incorporated fluorescent dye merely interacted with the blood cells and stayed inert in the plasma fraction.

### 2.4. In Vivo Fluorescence Imaging

For the detection of an accumulation in the deep tissue of the mice, the near-infrared fluorescent dye DiR was incorporated into the lipid phase of the nanoemulsions. The accumulation of the four different sized nanoemulsions was investigated in vivo by noninvasive fluorescence imaging in juvenile prepubescent (age: 3–4 weeks), adult (age: 12–39 weeks), and senescent (age: >48 weeks) female hairless immunocompetent SKH1-*Hr^hr^* Elite mice. Unfortunately, 2–3 min after the i.v. application of 100 µL NE25 nanoemulsion into the tail vein, the juvenile mouse suffered from nose- or throat bleeding as well as cramps. This mouse was redeemed immediately with CO_2_. The necropsy revealed no findings in the oral cavity, tongue, and teeth, but partially bleedings in the lungs and a blood clot in the trachea. Other studies reported toxicological effects for parenteral injected gold nanoparticles in the lung tissue of female mice especially for particle size of around 30 nm, whereas smaller-sized (1.5–13 nm) and bigger-sized (70 nm) nanoparticles showed less severe effects. Histological examination of the lung sections revealed several morphological changes such as enlarged airway cavities, epithelial hyperplasia, thinning of alveolar cell layer plus loss of elasticity, lesions, and emphysema leading to struggling for breath, fatigue, weight loss, and eventually death [5,51]. Since the suffering of that single mouse might be caused by the higher toxicity of the similar-sized nanoemulsion NE25, it was refrained to continue the animal trials with this formulation in juvenile mice. The adult and senescent mice showed no abnormalities after the application of the NE25 formulation, as well as all age groups with the other three formulations NE50, NE100, and NE150.

Figure 5 shows lateral, ventral, and dorsal in vivo fluorescence images of representative mice (one for each nanoemulsion and age group according to Equation (2) directly after the application (5–10 min) and 24 h after application). Already 5 to 10 min after the application, a slight accumulation of all nanoemulsions was observed in the liver in the ventral view, as shown by white arrows. This indicated a rapid hepatic uptake of the nanoemulsions from the circulating bloodstream as part of the hepatic clearance through the RES, involving the Kupffer cell-mediated phagocytosis in the liver, which was described as well for other nanoparticulate formulations within a few minutes or less than 1.5 h after parenteral applications [7,18,19,20]. The fluorescence signal intensity stated as radiant efficiency, increased significantly for the liver 24 h after application. Therefore, the highest radiant efficiencies were detected in the livers of the juvenile mice, followed by the adult mice with the formulations NE25 and NE50. The lowest radiant efficiency for the liver was found in senescent mice with the formulation NE150. However, these senescent mice with the NE150 formulation showed still a very broad distribution after 24 h spread over the whole body. Furthermore, an accumulation of all nanoemulsion formulation was detected in the ovaries of the adult mice 24 h after application, as shown with the red arrows. For the senescent group, solely the formulations NE25 and NE100 led to a visible accumulation in the ovaries, which were with a respective age of 64 and 49 weeks younger than the other senescent mice for the formulations NE50 and NE150 at respective ages of 85 and 80 weeks.

### 2.5. Ex Vivo Fluorescence Imaging and Biodistribution of the Nanoemulsions

For a more detailed overview of the accumulation of the four different sized nanoemulsions, ex vivo fluorescence imaging was conducted with 17 excised organs plus withdrawn blood by heart puncture of all sacrificed mice 24 h after i.v. injection. A region of interest (ROI) was drawn around the outlines of every single organ to determine their ARE. The PARE was then calculated for every single organ of each mouse according to (1) as well as the mean PARE for the five mice per group (nanoemulsion and age), which are provided in the Appendix A The mean PARE and the individual PARE values of each mouse are plotted as combined 3D bar chart and 3D scatter chart in Figure 6, respectively. Therefore, the nanoemulsion formulations NE25 (orange bars and scatter), NE50 (red), NE100 (blue), and NE150 (green) are plotted according to their age group on the *x*-axis. The ex vivo excised organs plus blood are plotted along the *y*-axis. The mean and individual PARE of the mice are plotted along the *z*-axis.

In addition to the overview of all excised organs, the mean PARE of the three organs with the highest accumulation (liver, spleen, and uterus + ovaries) plus four other organs with a mean PARE above 5% (heart, stomach, kidneys, and caecum) are plotted as contour plot against the particle size of the nanoemulsions and the age of the mice in Figure 7. Therefore, each black dot represents the exact age of the mice and the particle size of the nanoemulsions.

To verify the complete accumulation from the bloodstream into the organ tissue of the i.v. injected nanoemulsions, blood was withdrawn by heart puncture after scarification. The mean PARE of the blood was below 1% for each group, indicating that almost the whole formulation did not circulate in the bloodstream anymore. It either accumulated in the organ tissue within 24 h or was already excreted via the hepatic way.

As expected, the highest amount of the nanoemulsions accumulated in the liver tissue, as this organ is the main constituent in the RES accounting for up to 80–90% of its function [19,52,53]. Therefore, the particle size of the nanoemulsions had solely a marginal effect for the juvenile mice at the age of 3–4 weeks with a mean PARE between 60.8–65.6%, as shown in Figure 7a. The accumulation in the liver of adult and senescent mice however depended significantly on the particle size of the nanoemulsion, where the mean PARE decreased for both from around 66% to nearly 40% with increasing particle size. The clearance capacity of the RES in mice was found to decline significantly within the first 14 weeks after birth but then reached a constant level during further aging up to 128 weeks in mature and senescent mice [54,55], which may have caused the significant lower uptake of the bigger-sized nanoemulsions NE100 and NE150. For the smallest nanoemulsion NE25, the hepatic uptake might remain high, since the smaller the particle size, the better the uptake from the bloodstream into the cells for particles below 50 nm in size [6].

Besides the liver, the spleen and bone marrow play a vital role in the RES as well, containing reticular cells [54]. The spleen showed the second-highest accumulation of the four nanoemulsions. Therefore, the mean PARE was between 8.3–10.8% for the juvenile mice with an age of 3–4 weeks and showed a maximum peak of the mean PARE of 12.9% for adult mice at the age of 13 weeks with the 100 nm-sized nanoemulsion (Figure 7b). The accumulation in the spleen then declined with the age of the mice down to 7.3–8.8% indicating a decrease of the RES capacity as well with the age. The excised femur, knee, and tibia (not displayed as contour diagram) showed no dependency of neither the age nor the particle size of the nanoemulsions, as shown in Figure 6. Therefore, the small amount of accumulation with mean PARE between 3.0–4.6% was probably in the bone marrow as part of the RES. Overall the accumulation of the nanoemulsions decreased in the RES participating organs during the aging of the mice, which consequently led to a spreading of the accumulation over other non-RES organs.

A significant accumulation of the nanoemulsions was detected in the uterus and ovaries depending on their age and particle size (Figure 7c). The juvenile prepubescent mice younger than 4 weeks showed nearly no accumulation with a mean PARE between 1.2–1.7%. The apparent absence of the accumulation in the juvenile and immature mice probably derived from very low amounts of ovarian steroid hormones during that age and hence their non-fertility, since serum progesterone and estradiol levels significantly increase after 4–6 weeks of age with the onset of puberty [24,25]. The adult and hence fertile mice showed an increased accumulation of the nanoemulsions depending on particle size, where the highest mean PARE of 9.8% was detected for the biggest particle size of the NE150 at the age of 27 weeks. The mean PARE signal decreased with decreasing particle size down to 5.2% for the nanoemulsion NE25. With increasing age, the accumulation in the uterus and ovaries decreased to low mean PARE between 3.9–4-3% at the age of over 80 weeks. The decrease of accumulation correlated as well to the loss of fertility with increasing age, which is described with a gradual decrease of oocytes, shortening, and irregularities of the reproductive cycle until acyclicity and therewith cessation of steroid hormone cycling as well as a decline in litter size [21,23,30]. The slightly higher mean PARE 7.6% of the senescent mice with the NE100 nanoemulsion in Figure 6 was caused by the youngest age within the senescent group at an age of 49 weeks, compared to the other injections time points of 64.1 (NE25), 84.5 (NE50) and 80.4 weeks (NE150). However, this value fits in the decrease of accumulation over age, displayed in the contour plot (Figure 7c).

Independently of the age, the mean PARE of the heart increased from 2.1 to 5.9–6.7% with increasing particle size (Figure 7d), which was reported similarly for i.v. injected gold nanoparticles in female mice [5]. For the juvenile mice, there was nearly no accumulation detected in the stomach, kidneys, and caecum with mean PARE below 2.5%, as shown in Figure 7e–g. With aging and increasing particle size of the nanoemulsions, the mean PARE increased up to 6.4% in the stomach and kidneys and 5.2% in the caecum. Eventually, a small portion of the detected fluorescence signal in the stomach and caecum might derive from coprophagy. So, the mice might reuptake the fluorescence dye DiR into the gastrointestinal tract, which probably left the body already through the hepatic way. Coprophagy of mice was found to begin at the age of around 2.5 weeks after birth, a few days after beginning to excrete feces autonomously [56]. Hence, coprophagy began shortly before the i.v. injection of the nanoemulsions in the juvenile mice, which showed low fluorescence signals in the stomach and caecum with a PARE below 2.4%. The number of ingested fecal pellets was reported to peak at age of 5–6 weeks up to 13 pellets per day and then to decrease gradually with aging to 2.1 pellets per day at 78 weeks of age [56]. Therewith, the described increase of fluorescence signal in the stomach and caecum with aging stood in contrast to the decreasing coprophagy activity. Hence, coprophagy is unlikely the single factor for the increasing fluorescence signal in the stomach and caecum with aging, which might derive as well from the lower capacity of the RES in adult and senescent mice. So, in consequence, the accumulation of the nanoemulsion is distributed more widely across some other non-RES organs. All other excised organs (skin, lungs, s. c. fat, duodenum, pancreas and fat, thigh muscle, colon, bladder, and the brain) had mean PARE below 3.5% in declining order to the listing and therefore just a marginal accumulation of the nanoemulsions.

Ex vivo fluorescent images of the seven organs with the highest accumulations of the representative mice are displayed in Figure 8. The course of the previously discussed PARE data was confirmed by these ex vivo fluorescent images. Therefore, the radiant efficiency was mostly evenly distributed over the imaged organs with no distinctive hot spots, except for the uterus and ovaries. There, hot spots with high radiant efficiency were detected in the ovaries of all adult mice, as well for the senescent mice with NE25 and NE100 nanoemulsions, which were the younger ones of the senescent groups and probably did not lose their fertility as much as the other older senescent mice. In contrast, the uteri showed solely a moderate radiant efficiency for all age groups and nanoemulsions.

## 3. Materials and Methods

For the preparation of the nanoemulsions, PIONIER MCT (medium-chain triglyceride) and Kolliphor^®^ HS 15 (macrogol 15 hydroxystearate) were kindly provided by Hansen & Rosenthal KG (Hamburg, Germany) or BASF SE (Ludwigshafen, Germany), respectively. Sodium Chloride was purchased from Grüssing GmbH (Filsum, Germany) and the near-infrared fluorescent dye DiR (1,1’-dioctadecyl-3,3,3’,3’- tetramethylindotricarbocyanine iodide) was purchased from Invitrogen/Thermo Fisher Scientific Inc. (Carlsbad, CA, USA). Dulbecco’s modified eagle medium–high glucose with 4500 mg/L glucose, L-glutamine, sodium bicarbonate, with and without sodium pyruvate (DMEM w/NaP or DMEM *w*/*o* NaP, respectively), fetal calf serum (FCS), penicillin-streptomycin (P/S), Triton^TM^ X-100 and the fluorescent dye resazurin sodium salt for the cell toxicity assays, plus sodium citrate dehydrate and Dulbecco’s phosphate-buffered saline 10x were purchased from Sigma-Aldrich Chemie GmbH (Steinheim, Germany). Double distilled and 0.2 µm sterile filtered water was used in all experiments and analytics.

### 3.1. Preparation of Isotonic Nanoemulsions

Isotonic nanoemulsions with four different targeted particle sizes of 25 nm (NE25), 50 nm (NE50), 100 nm (NE100), and 150 nm (NE150) in diameter were prepared by a modified phase-inversion-based process, which is described thoroughly in a previous article [34]. For the noninvasive optical in vivo imaging, the fluorescent dye DiR was used to label the nanoemulsion droplets. Therefore, the solvent ethanol was evaporated from the DiR stock solution and the remaining dye was dissolved in MCT at a concentration of 0.1 mg/g. At 50 °C molten Kolliphor^®^ HS 15 and the loaded MCT were dispersed in aqueous NaCl solution under magnetic stirring at ~750 rpm using the compositions according to Table 1. The emulsion then was heated to 99 °C undergoing a phase inversion from an *o*/*w* to a *w*/*o* emulsion. The emulsion was cooled back into its phase inversion zone and shock diluted with the ice-cold water.

For the animal trials, the educts were sterilized before the nanoemulsion preparation. Therefore, MCT was sterilized by dry heat at 180 °C for 30 min, the Kolliphor^®^ HS 15 mixed with NaCl solution as well as the water for the shock dilution were autoclaved at 121 °C and 2 bar for 30 min including equilibration time. These nanoemulsions were then prepared as described beforehand, but aseptically in the Heraeus HERAsafe HS 12 laminar flow bench and finally sterile filtered with a 0.2 µm polyethersulfone (PES) filter into sterilized vials.

### 3.2. Particle Size Analysis, Zeta Potential, Osmolality, and pH Measurements

The particle diameters and zeta potentials were measured with the Malvern Instruments Zetasizer Nano ZS. For the particle diameter measurements, each individually produced batch was diluted at 1:100 in water and measured in triplicate with 15 runs each at 25 °C in backscattering mode. The zeta potential was measured in triplicate with each batch diluted at 1:10 in 0.1x PBS buffer with pH 7.4 at 25 °C with 10 to 50 runs per measurement. The osmolality of each individually produced batch was determined as well in triplicate with the KNAUER Semi-Mikro Osmometer. The pH values were determined with single-point measurements of each batch with the Knick Portamess^®^ 913 pH meter.

### 3.3. Determination of the IC50 on 3T3 and NHDF Fibroblasts

Toxicity assays with 3T3 and NHDF fibroblasts were carried out in triplicate per cell line at eight replicates per run, aseptically in the Heraeus HERAsafe HS 12 laminar flow bench by seeding approximately 20,000 NHDF cells or 10,000 3T3 cells in 11 of the 12 columns of the 96-well plates. The seeded cells were then grown in the Heraeus HERAcell incubator for 24 h at 37 °C and 5% CO_2_ in 100 µL DMEM w/NaP for 3T3 fibroblasts or DMEM *w*/*o* NaP for NHDF fibroblasts, both media with additionally 10 vol.% FCS and 1 vol.% P/S. Therefore, the first of the 12 columns was left without cells, hence filled solely with a cell culture medium for the determination of the blank. The cells in the second column were left untreated by adding 100 µL cell culture medium as a negative control of viable cells and the third column was treated by adding 100 µL with 0.05 vol.% Triton^TM^ X-100 solution (final concentration 0.025 vol.%) as positive control of fully inhibited cells. For the remaining nine columns, each nanoemulsion was sterile filtered with 0.2 µm PES filter and then diluted up to 10 times at ratios of 1:2.35 with the corresponding cell culture medium. A total of 100 µL of the different diluted nanoemulsions were added to the remaining nine columns with NHDF or 3T3 cells and the well plates were incubated for another 4 or 24 h. The cell viability was determined by a resazurin reduction assay. Therefore, 20 µL of 440 µM resazurin solution was added and mixed thoroughly with the cell culture medium by carefully withdrawing the medium back and forth into the pipette tips. The mixture was incubated for 2 h and the fluorescence intensity was determined with the BioTek^®^ Instruments Cytation^TM^ 5 imaging reader using the RFP 531(excitation)/593(emission) filter set. The cell viability was calculated as the percentage of the negative controls (untreated cells) after subtraction of the blank (only medium without cells). The IC50 was then calculated by linear interpolation.

### 3.4. Investigation of the Nanoemulsion Interaction with Blood Cells

Whole mouse blood of untreated mice was stabilized against coagulation by adding 0.109 M sodium citrate at a ratio 1:9. A total of 100 µL of the stabilized whole blood was incubated for 4 h with 50 µL of each nanoemulsion at room temperature. For the separation of the blood cell fraction from the plasma, the incubated blood was centrifuged at 3000 rpm for 15 min with the Eppendorf^TM^ MiniSpin^®^ Centrifuge and the plasma was pipetted off. For washing, the pellets with the blood cell fraction were repeatedly re-dispersed in 1× PBS and centrifuged three times in total. Both plasma and blood cell fractions, as well as untreated whole blood were then transferred into a 96-well plate and fluorescence imaging was performed of the well plates as described in Section 3.6. The PARE of both fractions was determined after the subtraction of the untreated whole blood ARE as blank, according to Equation (1) in Section 3.7.

### 3.5. Animal Handling

All in vivo protocols were approved by local authorities of Saxony-Anhalt, Germany, under file number 42502-2-1456 MLU and complied with the guidelines of the Federation for Laboratory Animal Science Associations (FELASA). The in vivo studies were performed in juvenile (age: 3–4 weeks), adult (age: 12–39 weeks), and senescent (age: >48 weeks) female hairless immunocompetent SKH1-*Hr^hr^* Elite mice (mouse model of Charles River, Sulzfeld, Germany), as illustrated in Figure 9a with changes of mouse fertility. The mice were bred by the center for basic research (ZMG) of the Martin Luther University Halle-Wittenberg. The mice were housed in windowless and fully automated air-conditioned rooms with barrier-maintained mouse colonies in closed cage systems type II long IVC with a maximum of five mice per cage according to the EG guideline 86/609/EWG and GV-SOLAS guidelines. The light and temperature regime was controlled automatically and the feeding followed the special diet for lab mice. Mice had access to water and food ad libidum. 

The general experimental procedure of the animal trails is shown in Figure 9b. Five mice for each nanoemulsion formulation and age group were slowly injected i.v. with a 30 G cannula into their tail veins using a restrainer, darkened with a red foil. Thereby, 100 µL nanoemulsion was applied for the juvenile (mean bodyweight: 15.2 ± 3.4 g) and 200 µL nanoemulsion for the adult (mean bodyweight: 30.0 ± 3.8 g) and senescent mice (mean bodyweight: 37.6 ± 4.7 g), resulting in a mean dose of 6.3 ± 1.3 µL/g bodyweight. For in vivo fluorescence imaging directly after injection (5–10 min, so that the bleeding of the injection site stopped) and after 24 h, the mice were anesthetized with the Caliper LifeSciences XGI-8 Gas Anesthesia System and MIDMARK^®^ Matrx VIP 3000^®^ Isoflurane Vaporizer at a 1–2% isoflurane-oxygen mixture at ~3.0 L/min O_2_ initial flow in the induction chamber and ~1.5 L/min steady flow during fluorescence imaging, both flows at 21.1 °C and atmospheric pressure. During fluorescence imaging, the mice were placed on a 37 °C temperature-controlled stage, preventing a body temperature decreasing in the mice. The mice were sacrificed by cervical dislocation 24 h after injection and for further ex vivo analysis, 17 organs and whole blood samples were withdrawn during necropsy.

### 3.6. Fluorescence Imaging

Both in vivo and ex vivo fluorescence imaging was performed with the PerkinElmer^®^ IVIS^®^ SPECTRUM fluorescence imager using the epi-illumination mode with a filter pair at auto exposure. Thereby, the excitation filter was set to 745 nm with 20 nm bandwidth and emission filter to 800 nm with 30 nm bandwidth to detect the near-infrared dye DiR. For the in vivo fluorescence imaging of every single mouse directly after injection and after 24 h, the field of view was set to D using the mouse as the set subject. The field of view was set to C using the well plate as the set subject for the ex vivo fluorescence imaging of the different excised organs displayed in a 12-well plate.

### 3.7. Image and Data Processing of the Fluorescence Images

For the image and data processing of the fluorescence imaging, the PerkinElmer^®^ Living Image^®^ 4.7.3 Software was used. For both the in vivo and ex vivo fluorescence image adjustments, the brightness was set to 100, contrast to 1.5, and opacity to 60 using the “rainbow2” color table in reverse at a logarithmic scale. The binning was set to 2 and the smoothing to “none”. For the setting of the color scale, in vivo and ex vivo fluorescence images were taken of two untreated mice. The maximal radiant efficiencies of the in vivo, as well as ex vivo fluorescence images were determined. Both values were below 3.0 × 10^7^ (p/sec/cm^2^/sr)/(µW/cm^2^). Thus, setting the minimum of the color scale to 3.0 × 10^7^ (p/sec/cm^2^/sr)/(µW/cm^2^) did cut off the autofluorescence of the mouse tissue in vivo as well as ex vivo. The maximum of the color scale was set just right above the maximal detected radiant efficiency of all taken fluorescence images to 3.0 × 10^7^ or 5.0 × 10^9^ (p/sec/cm^2^/sr)/(µW/cm^2^) for the in vivo and ex vivo fluorescence images, respectively.

For data processing of the ex vivo fluorescence images, the intensity signal of each excised organ was measured as ARE. Therefore, a ROI was drawn around each single excised organ using the “auto 1” function, automatically drawing one ROI around the selected organ using the threshold to reach as close to the outline of the organ. In case the “auto 1” function was not applicable at organs with very low fluorescence signal, the ROI was drawn manually using the “free draw” function. The surface intensity signals of all organs then were measured as ARE, giving the sum of the fluorescence emission radiance from each pixel inside the ROI per number of pixels (photons/sec/cm^2^/sr) divided through the incident excitation intensity (µW/cm^2^). For comparison, the *PARE_m,n_* of each single organ n of a mouse m was expressed as a portion of the sum of all measured *ARE_m,n_* signals of the 17 excised organs + blood (*n* = 1…18) of that single mouse:(1)$$PARE_{m,n}=\frac{ARE_{m,n}}{\sum_{n=1}^{18}ARE_{m,n}}$$

The mean PARE was calculated for the 3D bar chart and logarithmic contour plots in Figure 6 and Figure 7 as follows:
(2)$$\emptyset PARE_{n}=\frac{\sum_{m=1}^{5}PARE_{m,n}}{5}$$

For the depiction of the fluorescence images in Figure 5 and Figure 8, representative mice were selected, which deviated least from the mean fluorescence signals between the five mice of each investigated group (age and nanoemulsion particle size). Therefore, the sum of the absolute values from the difference between the mean ∅*PARE_n_* of all five mice and the *PARE_m,n_* of the single mouse was calculated for each mouse in that group, giving the deviation *Dev_m_* of each mouse to the mean of the five mice:
(3)$$Dev_{m}=\sum_{n=1}^{18}\left|\emptyset PARE_{n}-PARE_{m,n}\right|$$

The mouse with the smallest *Dev_m_* was then chosen as the representative mouse.

## 4. Conclusions

This in vivo study showed a significant accumulation of nanoemulsions in the ovaries of female mice, which depended on both the particle size and the age of the mice. For juvenile and immature mice younger than 4 weeks, nearly no accumulation was observed in the ovaries, mostly accumulating in the RES organs liver, and spleen. After puberty and hence in the most fertile life period, the ovarian accumulation of the nanoemulsions increased to a remarkable extent in adult mice at an age of 13–39 weeks. Therefore, particle size played a significant role in ovarian accumulation. Within the investigated size range and the chosen nanoemulsion system, the following observation was made: the bigger the nanoemulsion particle size, the higher was the accumulation in the ovaries with up to nearly 10% PARE of all excised organs. To reduce unintended accumulation and therewith possible side effects in the ovaries during the fertile lifetime period, a reduction of particle size is recommended for the chosen nanoemulsion system, since the small-sized nanoemulsions reduced the amount of ovarian accumulation to nearly half in comparison to the 150 nm-sized nanoemulsions. With increasing age and hence decreasing fertility, unintended ovarian accumulation of all different sized nanoemulsions declined to low levels with increasing senescence and therefore reduced the possibility of side effects such as nanotoxicity.

In case a drug delivery into ovarian tissue is desired, for example for ovarian cancer treatment, increasing the particle size of this nanosized DDS might increase treatment success, since the accumulation in ovarian tissue was nearly doubled by increasing the particle size of the nanoemulsions during the fertile lifetime period of the mice.

## Figures and Tables

**Figure 1 ijms-22-08283-f001:**
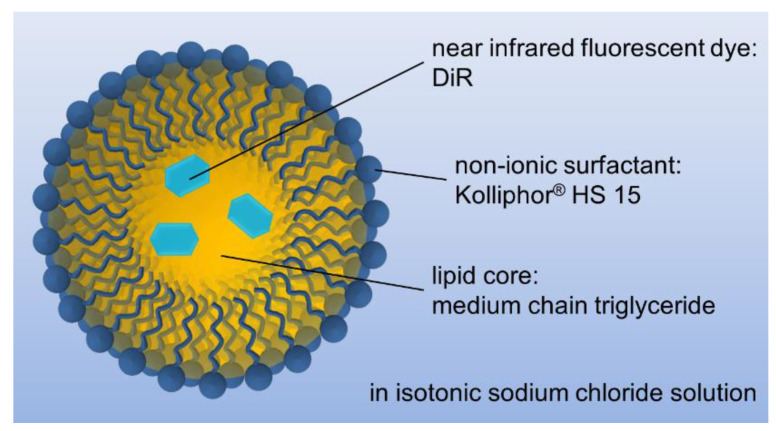
Composition of the chosen nanoemulsion for this in vivo study.

**Figure 2 ijms-22-08283-f002:**
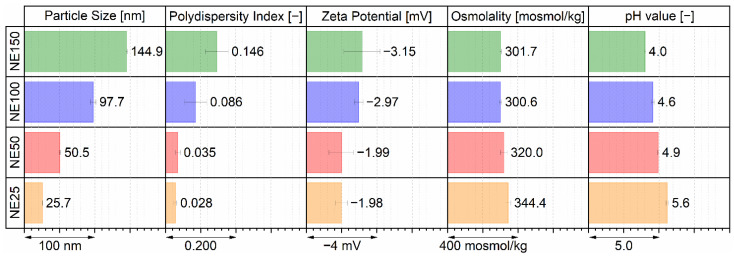
Physicochemical properties of the four different sized nanoemulsions, determined by three individual produced batches.

**Figure 3 ijms-22-08283-f003:**
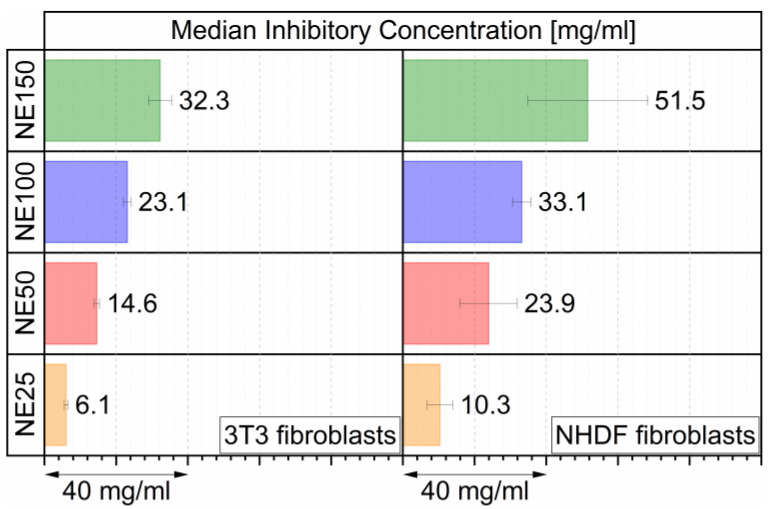
IC50 as total mass nanoemulsion (DiR loaded MCT, Kolliphor^®^ HS 15 + aqueous phase) per ml cell culture media on 3T3 and NHDF fibroblasts after 24 h of incubation, determined by three individual incubated batches (eight replicates per run).

**Figure 4 ijms-22-08283-f004:**
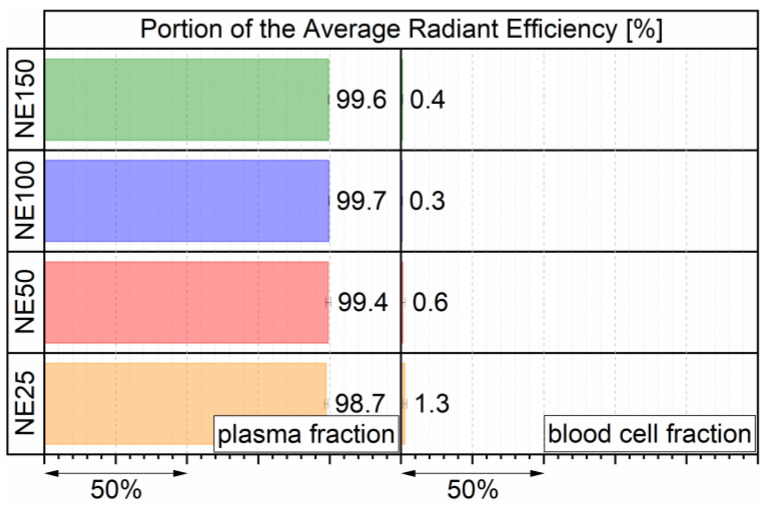
PARE in plasma and blood cell fractions after 4 h of incubation, determined by three individual incubated batches.

**Figure 5 ijms-22-08283-f005:**
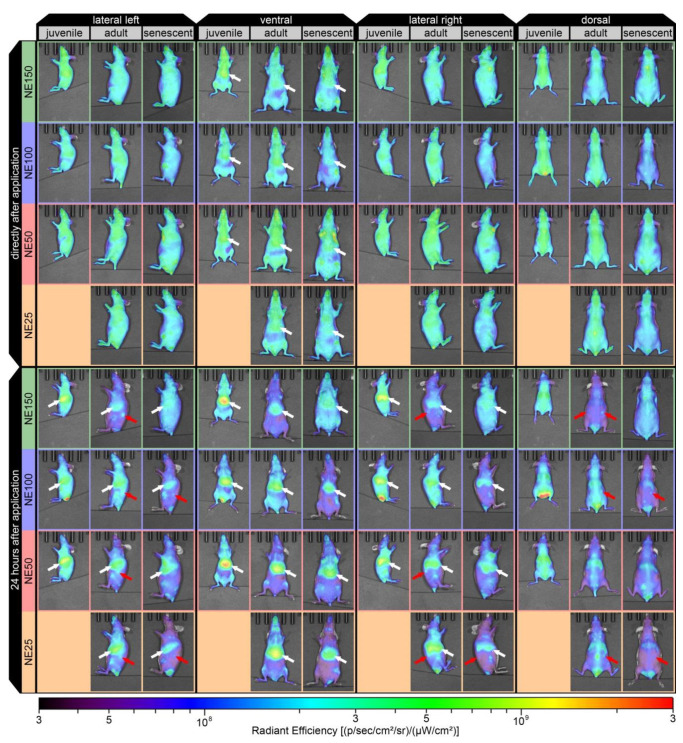
Noninvasive in vivo fluorescence images of representative mice directly after (5–10 min) and 24 h after i.v. injection in the tail vein of the four nanoemulsions in juvenile prepubescent (age 3–4 weeks), adult (age 12–39 weeks), and senescent mice (age > 48 weeks).

**Figure 6 ijms-22-08283-f006:**
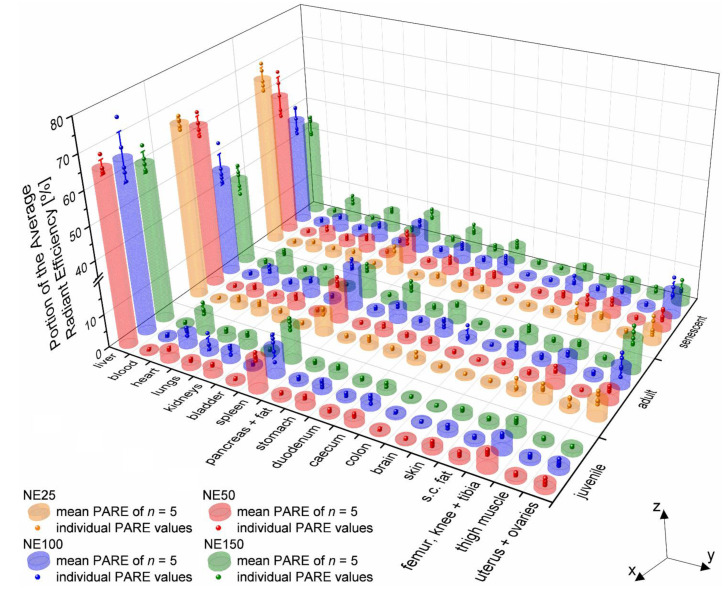
Combined 3D bar chart (mean PARE of the five mice per group) and scatter chart (individual PARE values of each mouse) of the ex vivo excised organs and blood, 24 h after i.v. injection of the four nanoemulsions in juvenile prepubescent (age 3–4 weeks), adult (age 12–39 weeks), and senescent mice (age > 48 weeks).

**Figure 7 ijms-22-08283-f007:**
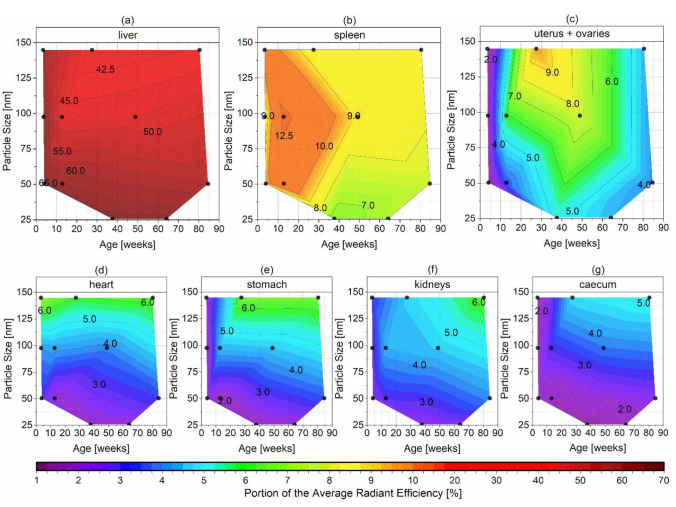
Logarithmic contour plots of the mean PARE for the organs with the highest accumulations ((**a**) liver, (**b**) spleen, (**c**) uterus + ovaries, (**d**) heart, (**e**) stomach, (**f**) kidneys, and (**g**) caecum) plotted against the particle size of the i.v. injected nanoemulsions and the exact age of the mice displayed as ●.

**Figure 8 ijms-22-08283-f008:**
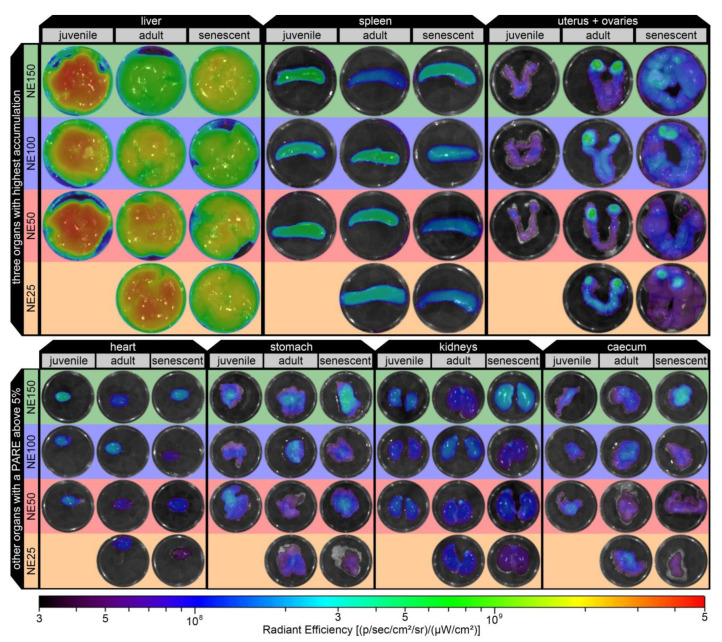
Ex vivo fluorescence images of the excised organs with the highest accumulation (liver, spleen, uterus +ovaries, heart, stomach, kidneys, and caecum) of the representative mice.

**Figure 9 ijms-22-08283-f009:**
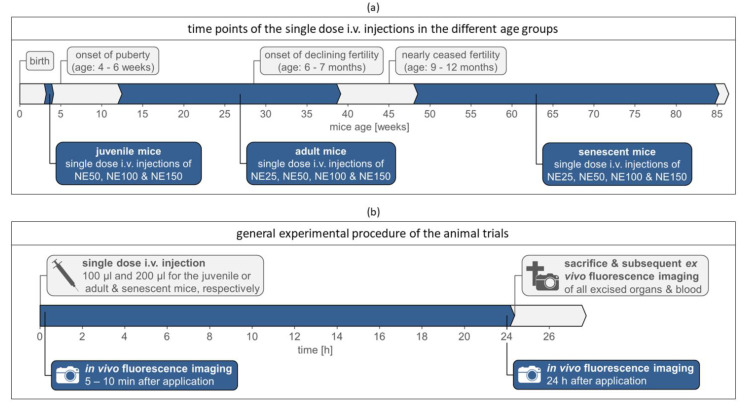
Experimental design of the animal trials: (**a**) time points of the single-dose i.v. injections with the nanoemulsions in the age groups juvenile, adult, and senescent, including the onset of puberty [24,25,26], the onset of declining fertility, and nearly ceased fertility of female mice [21,23,28,29,30,31,32,33]; (**b**) general experimental procedure of the animal trials.

**Table 1 ijms-22-08283-t001:** Composition of the isotonic nanoemulsions.

Compounds in wt.%	NE25	NE50	NE100	NE150
DiR loaded MCT ^1^	8.0	8.0	8.0	8.0
Kolliphor^®^ HS 15	20.0	8.8	5.3	4.0
NaCl solution	25.3 ^2^	23.2 ^3^	13.3 ^4^	8.0 ^5^
ice-cold water	46.7	60.0	73.3	80.0

^1^ DiR loaded MCT at a concentration of 0.1 mg/g; NaCl solutions at salinities of ^2^ 0.24 wt.%, ^3^ 2.35 wt.%, ^4^ 4.96 wt.%, and ^5^ 8.78 wt.%

## Data Availability

Not applicable.

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
