# Peer review of "Ovarian Accumulation of Nanoemulsions: Impact of Mice Age and Particle Size"

_ijms, 2021, doi:10.3390/ijms22158283_

Round 1
Reviewer 1 Report
Title
The title should be revised. In my opinion, the use of the word “unintended” in the title does not make sense given that in the present study the authors (intentionally) explored the role of particle size and female reproductive age in the biodistribution, including ovarian accumulation, of nanoemulsions (NE). In fact, the closing sentence of the Conclusion section states that: “In case a drug delivery into ovarian tissue is desired, for example for ovarian cancer treatment, increasing the particle size of this nanosized DDS might increase treatment success, since the accumulation in ovarian tissue was nearly doubled by increasing the particle size of the nanoemulsions during the fertile lifetime period of the mice.”
Doses tested
It is not clear from the text the exact doses of the different NE injected in mice. The description in the Materials and Methods section (page 14, lines 441-443) - “Thereby, 100 μl nanoemulsion was applied for the juvenile and 200 μl nanoemulsion for the adult and senescent mice, since the body weights of the juvenile mice were much lower than the body weights of the adult and senescent mice.”- gives the idea that dosing has not been adjusted based on body weight of individual mice as one would expect in a biodistribution study. The authors must clarify this important point. Moreover, the rational for the choice of the administered doses must be indicated and how they relate to a real-world application.
In fact, dosing variation might account for high variation in NE biodistribution as determined by the detected fluorescence signals. In this regard, the authors state that a selection of the most representative mouse between the five has been made (page 15, lines 489-494). The authors must report the mean and not the values for the most representative mouse. The authors should also report the obtained values in terms of % of injected dose.
Table 1
Do authors really refer to cell culture media composition?
Table 2 should be removed, and the final concentration of each cell culture component indicated in the text.
Why have the authors chosen to test the cytotoxicity of the NE in fibroblast and not in ovarian cell lines? What is the relation between in vitro and in vivo tested doses?
How many independent experiments have been performed? How many replicates per experiment have been run?
Figure 6
The authors must present the mean and not individual mice values. In my opinion, a Table would be more informative to the reader than a 3D bar chart.
Regarding the in vitro cytotoxicity study, the authors must indicate the tested concentrations (in mass/volume or mass/area.
Non-linear regression (dose-response curves) rather than linear interpolation is the best choice for determining IC50 values.
Author Response
Reviewer 1: Title
The title should be revised. In my opinion, the use of the word “unintended” in the title does not make sense given that in the present study the authors (intentionally) explored the role of particle size and female reproductive age in the biodistribution, including ovarian accumulation, of nanoemulsions (NE). In fact, the closing sentence of the Conclusion section states that: “In case a drug delivery into ovarian tissue is desired, for example for ovarian cancer treatment, increasing the particle size of this nanosized DDS might increase treatment success, since the accumulation in ovarian tissue was nearly doubled by increasing the particle size of the nanoemulsions during the fertile lifetime period of the mice.”
Author's Response: The title is changed and the word “unintended” deleted.
Reviewer 1: Doses tested
It is not clear from the text the exact doses of the different NE injected in mice. The description in the Materials and Methods section (page 14, lines 441-443) - “Thereby, 100 μl nanoemulsion was applied for the juvenile and 200 μl nanoemulsion for the adult and senescent mice, since the body weights of the juvenile mice were much lower than the body weights of the adult and senescent mice.”- gives the idea that dosing has not been adjusted based on body weight of individual mice as one would expect in a biodistribution study. The authors must clarify this important point. Moreover, the rational for the choice of the administered doses must be indicated and how they relate to a real-world application.
Author's Response: The mean bodyweight of the juvenile mice was 15.2 ± 3.4 g, the body weight of the adult mice was 30.0 ± 3.8g and the body weight of the senescent mice was 37.6 ± 4.7 g. Since the bodyweight of the juvenile mice was half the mass of the adult mice, the dose was halved for this age group. Overall, this results in a mean dosis of 6.3 ± 1.3 µl/g bodyweight. The mean bodyweights of the age groups and the mean dosis are now added in the corresponding sentence.
Since we focused this study on the detectability of the accumulation of the carrier system by fluorescence imaging, we adjusted the dosis to achieve optimal detection of the fluorescent dye DiR by in vivo and ex vivo fluorescence imaging. In the case, that the nanocarrier would be loaded with a specific active pharmaceutical ingredient, the dosis should be adjusted according to the drug load in the nanoemulsions.
Reviewer 1: In fact, dosing variation might account for high variation in NE biodistribution as determined by the detected fluorescence signals. In this regard, the authors state that a selection of the most representative mouse between the five has been made (page 15, lines 489-494). The authors must report the mean and not the values for the most representative mouse. The authors should also report the obtained values in terms of % of injected dose.
Author's Response: The selection of the most representative mouse was done only for the fluorescence images in figure 5 and figure 8. The 3D bar chart in figure 6 and the logarithmic contour plots in figure 7 do represent the mean PARE values as stated in their captions now. For clarification, the chapter was rewritten accordingly.
Reviewer 1: Table 1
Do authors really refer to cell culture media composition?
Author's Response: The caption of the table is corrected.
Reviewer 1: Table 2 should be removed, and the final concentration of each cell culture component indicated in the text.
Author's Response: Table 2 is removed and the final concentrations of both cell culture components are indicated in the text now.
Reviewer 1: Why have the authors chosen to test the cytotoxicity of the NE in fibroblast and not in ovarian cell lines? What is the relation between in vitro and in vivo tested doses?
Author's Response: Due to the systemic distribution of the nanoemulsions after the intravenous application and following the accumulation in many different tissues e.g. the liver, spleen, ovaries etc., we chose an unspecific cellular model rather than one of a specific ovarian cell line, representing only one of the many different ovarian tissue types. Therefore, we found the 3T3 and NHDF fibroblasts suitable to investigate the situation in healthy and unspecific tissue. We did not choose any carcinoma cells lines, since those cell lines may not represent the properties of healthy tissue, being possibly more resistant against the investigated nanoemulsions. For the future, if the ovarian accumulation of the drug delivery systems clearly points out in one specific ovarian tissue, the cellular toxicity should be investigated additionally with the corresponding ovarian cell line.
The in vitro data served as a reference point, since we focused on the in vivo monitoring of the biodistribution of the nanoemulsions. Therefore, we applied a maximal possible dosis to achieve optimal detection sensitivity of the fluorescence signal being limited with the fluorescence dye load in the MCT droplets because of quenching effects. On the basis of a previous study with lipid nanocapsules by Hirsjärvi, et al. (see page 596, 1st paragraph, line 3), we decided to inject 200 µl as well intravenously into the tail vein.
Hirsjärvi, S.; Sancey, L.; Dufort, S.; Belloche, C.; Vanpouille-Box, C.; Garcion, E.; Coll, J. L.; Hindré, F.; Benoît, J. P. Effect of Particle Size on the Biodistribution of Lipid Nanocapsules: Comparison between Nuclear and Fluorescence Imaging and Counting. Int. J. Pharm. 2013, 453 (2), 594–600. https://doi.org/10.1016/j.ijpharm.2013.05.057.
Reviewer 1: How many independent experiments have been performed? How many replicates per experiment have been run?
Author's Response: Per cell line and nanoemulsion formulation of the four different particle sizes, three independent experiments were performed with eight replicates per run, which is now stated in the method chapter and caption of figure 3.
Reviewer 1: Figure 6
The authors must present the mean and not individual mice values. In my opinion, a Table would be more informative to the reader than a 3D bar chart.
Author's Response: The bars in figure 6 represent the mean PARE plus standard deviation of all five mice per group. The scatters represent the individual values of each single mouse. For better understanding, the legend of the figure plus the caption was edited. In our opinion, a single table of the nearly 200 mean PARE values (without any standard deviation values) is probably confusing and complex to read and compare between the different mice groups (age and particle size). Hence, we decided to create the 3D bar chart to give a quick overview of the PARE values of all 17 excised organs plus the withdrawn blood of the eleven different mice groups. For those interested in the exact mean PARE values and their standard deviations, we added the table S1 as supplementary material.
Reviewer 1: Regarding the in vitro cytotoxicity study, the authors must indicate the tested concentrations (in mass/volume or mass/area.
Author's Response: The concentration of the IC50 values refer to the total mass nanoemulsion (DiR loaded MCT, Kolliphor® HS 15 + aqueous phase) per ml cell culture media. The caption of figure 3 is now changed to: “Median inhibitory concentration as total mass nanoemulsion (DiR loaded MCT, Kolliphor® HS 15 + aqueous phase) per ml cell culture media on 3T3 and NHDF fibroblasts after 24 h incubation, determined by three individual incubated batches (eight replicates per run).” For further information of the tested concentrations, the following dose response curves are now attached as supplementary figure S1.
Reviewer 1: Figure S1: Dose response curves of the cell viability over the total mass nanoemulsion (DiR loaded MCT, Kolliphor® HS 15 + aqueous phase) per ml cell culture media on (a) 3T3 and (b) NHDF fibroblasts after 24 h of incubation, determined by three individual incubated batches (eight replicates per run); the inlet graphs show the corresponding negative and positive controls with the untreated or TritonTM X-100 treated cells, respectively.
Non-linear regression (dose-response curves) rather than linear interpolation is the best choice for determining IC50 values.
Author's Response: The determined dose response curves did not show a usual sigmoidal curve, as illustrated in the following picture with the NE150 on 3T3 fibroblasts as an example. The dose response curve is overlapped by growth promoting effects at low nanoemulsion concentrations and with increasing concentrations cytotoxic effects became dominant. Therefore, the sigmoidal dose response fit did not seem to be suitable to determine the IC50 values correctly (see following 3T3 plot). Thus, we decided to calculate the IC50 by linear interpolation between the two points above and below 50 % cell viability.
Reviewer 2 Report
General comments
In the manuscript “Unintended ovarian accumulation of nanoemulsions: impact of mice age and particle size” the authors investigated both the influence of particle size and aging effects onto the accumulation of nanoparticles in ovarian tissue.
The topic is of significant relevance due to its impact on human health, especially reproductive health, so this manuscript could provide important reference information that are of great interest.
The manuscript is well written, the experimental design appears remarkable but it is not clearly illustrated. The figures are appropriate. In addition, the authors have consulted a sufficient number of scientific papers (56).
Specific comments
It is not clear the nanoemulsion chemical composition. Please add a detailed description in Material and Methods section
Author Response
Reviewer 2: In the manuscript “Unintended ovarian accumulation of nanoemulsions: impact of mice age and particle size” the authors investigated both the influence of particle size and aging effects onto the accumulation of nanoparticles in ovarian tissue.
The topic is of significant relevance due to its impact on human health, especially reproductive health, so this manuscript could provide important reference information that are of great interest.
The manuscript is well written, the experimental design appears remarkable but it is not clearly illustrated. The figures are appropriate. In addition, the authors have consulted a sufficient number of scientific papers (56).
Author's Response: For better understanding of the experimental design of the animal trails, we added an illustration in figure 9 depicting (a) the time points of the i.v. application in the different age groups in relation to the onset of puberty, decline of fertility and nearly ceased fertility and (b) the general experimental procedure of our animal trails.
Reviewer 2: It is not clear the nanoemulsion chemical composition. Please add a detailed description in Material and Methods section
Author's Response: Unfortunately, the caption of table 1 was labeled wrong and is now changed to “Composition of the isotonic nanoemulsions”, which lists the mass shares off each compound used for the four different nanoemulsions. We think that the figure 1 and table 1 are sufficient to provide the information of the chemical composition of the nanoemulsions.
Reviewer 3 Report
The manuscript by Busmann et al. describes the synthesis of four types of nanoemulsion of different sizes. The biodistribution of the nanoemulsions was investigated using mice of different ages, from juvenile prepubescent to senescent mice. Interestingly, the ovarian accumulation is influenced by the particle size, but also by the age. Although the results are interesting, the manuscript lacks important information that should be included in order for it to be accepted.
1- DLS has been measured in water and Z potential values are close to zero, which generally indicates a low stability at that pH (pH 7.4 in this case). Are the nanoemulsions stable at physiological buffers such as PBS? A stability test in PBS and 37ºC should be included.
2- Biodistribution measurements are based on fluorescent signals. Have the authors quantified the amount of fluorophore entrapped in each type of nanoemulsion? In order to directly compare all the groups, the amount of dye should be similar.
3- Similarly, and in order to compare the results obtained in vivo: was the dose of nanoemulsion injected per gram of animal the same for all formulations and groups? How did the authors calculate the concentration of nanoemulsion in each group? . How was the nanoemulsion content calculated?
4- Are the doses injected into the animals related to the IC50 obtained in in vitro culture?
5- Control mice without receiving nanoemulsions should be included to check the possible autofluorescence.
Author Response
Reviewer 3: The manuscript by Busmann et al. describes the synthesis of four types of nanoemulsion of different sizes. The biodistribution of the nanoemulsions was investigated using mice of different ages, from juvenile prepubescent to senescent mice. Interestingly, the ovarian accumulation is influenced by the particle size, but also by the age. Although the results are interesting, the manuscript lacks important information that should be included in order for it to be accepted.
1- DLS has been measured in water and Z potential values are close to zero, which generally indicates a low stability at that pH (pH 7.4 in this case). Are the nanoemulsions stable at physiological buffers such as PBS? A stability test in PBS and 37ºC should be included.
Author's Response: Although the zeta potential is nearly neutral, the nanoemulsions are stabilized against coagulation of the emulsion droplets by steric stabilization through the long chains of the non-ionic surfactant Kolliphor HS 15. Previous research showed a storage stability of the nanoemulsions from 5°C up to 40 °C with the recommended storage conditions according to the ICH guidelines Q1A for at least of 4 weeks for the NE25 and 8 weeks for the NE50, NE100 and NE150, as you can see in excerpt of our previous article:
Busmann, E. F.; García Martínez, D.; Lucas, H.; Mäder, K. Phase Inversion-Based Nanoemulsions of Medium Chain Triglyceride as Potential Drug Delivery System for Parenteral Applications. Beilstein J. Nanotechnol. 2020, 11, 213–224. https://doi.org/10.3762/bjnano.11.16.
Since the measured zeta-potential of the nanoemulsions diluted solely 1:10 in distilled water (not published in the article) were nearly neutral for all the four nanoemulsions as well, we think that the stability in PBS will be equivalent to the measured storage stability data. We hope the data of the storage stability is sufficient to clear your concerns of the nanoemulsion stability.
Reviewer 3: 2- Biodistribution measurements are based on fluorescencent signals. Have the authors quantified the amount of fluorophore entrapped in each type of nanoemulsion? In order to directly compare all the groups, the amount of dye should be similar.
Author's Response: Each nanoemulsion was produced using the same mass share of 8 wt.% DiR loaded MCT, as stated in table 1. Thereby the concentration of DiR in MCT was 0.1 mg/g, leading to the same final DiR concentrations of 0.008 mg/g for all four nanoemulsions NE25, NE50, NE100 and NE150. Since the fluorescent dye is very lipophilic with a reported logP of 17.4, it can be assumed, that the dye is completely entrapped in the lipid core of the nanoemulsion droplets.
Reviewer 3: 3- Similarly, and in order to compare the results obtained in vivo: was the dose of nanoemulsion injected per gram of animal the same for all formulations and groups? How did the authors calculate the concentration of nanoemulsion in each group? How was the nanoemulsion content calculated?
Author's Response: The mean bodyweight of the juvenile mice was 15.2 ± 3.4 g, the body weight of the adult mice was 30.0 ± 3.8g and the body weight of the senescent mice was 37.6 ± 4.7 g. Since the mean bodyweight of the juvenile mice was half the mass of the adult mice, the dose was halved for this age group. Overall, this results in a mean doses of 6.3 ± 1.3 µl/g bodyweight. The mean bodyweights of the age groups and the mean doses are now added in the corresponding sentence.
Reviewer 3: 4- Are the doses injected into the animals related to the IC50 obtained in in vitro culture?
Author's Response: The in vitro data served as a reference point, since we focused on the in vivo monitoring of the biodistribution of the nanoemulsions. Therefore, we applied a maximal possible dosis to achieve optimal detection sensitivity of the fluorescence signal being limited with the fluorescence dye load in the MCT droplets because of quenching effects. On the basis of a previous study with lipid nanocapsules by Hirsjärvi, et al. (see page 596, 1st paragraph, line 3), we decided to inject 200 µl as well intravenously into the tail vein.
Hirsjärvi, S.; Sancey, L.; Dufort, S.; Belloche, C.; Vanpouille-Box, C.; Garcion, E.; Coll, J. L.; Hindré, F.; Benoît, J. P. Effect of Particle Size on the Biodistribution of Lipid Nanocapsules: Comparison between Nuclear and Fluorescence Imaging and Counting. Int. J. Pharm. 2013, 453 (2), 594–600. https://doi.org/10.1016/j.ijpharm.2013.05.057.
Reviewer 3: 5- Control mice without receiving nanoemulsions should be included to check the possible autofluorescence.
Author's Response: As stated in chapter 3.7. “Image and data processing of the fluorescence images” the autofluorescence of the two untreated mice was detected below an radiant efficiency of 3.0 x 107 (p/sec/cm²/sr)/(µW/cm²). Thus, the minimum of the colour bar was set to this value to cut off the autofluorescence. Adding the fluorescence images into the figures of the untreated mice would probably lead to a lot smaller images in the figures 5 and 8 and therewith compromise the visibility of the other fluorescence images with the treated mice without adding significant information for the reader.
Round 2
Reviewer 2 Report
The changes I requested for the manuscript were all satisfactory. I think that, in this revised form, the paper makes a significant contribution to the field. Then, I recommend it for publication in International Journal of Molecular Sciences.
Author Response
Thank you very much for your kind comments.
Reviewer 3 Report
The manuscript can be accepted in its present form
Author Response

(The authors gave the same response as above.)
